# Preliminary Study on Type I Interferon as a Mucosal Adjuvant for Human Respiratory Syncytial Virus F Protein

**DOI:** 10.3390/vaccines12111297

**Published:** 2024-11-20

**Authors:** Hongqiao Hu, Li Zhang, Lei Cao, Jie Jiang, Yuqing Shi, Hong Guo, Yang Wang, Hai Li, Yan Zhang

**Affiliations:** 1NHC Key Laboratory of Medical Virology and Viral Diseases, National Institute for Viral Disease Control and Prevention, Chinese Center for Disease Control and Prevention, WHO WPRO Regional Reference Measles/Rubella Laboratory, Beijing 102206, China; huhq046@163.com (H.H.); zhangli2861264014@163.com (L.Z.); clsunrise@126.com (L.C.); jiejiang0317@163.com (J.J.); shiyuqing1997@yeah.net (Y.S.); gh199408@163.com (H.G.); 2Shandong First Medical University (Shandong Academy of Medical Sciences), Jinan 250117, China; 3Beijing Jishuitan Hospital, Beijing 100035, China; bucmwy@163.com

**Keywords:** human respiratory syncytial virus, type I interferon, mucosal adjuvant, protective effects

## Abstract

**Background:**Human respiratory syncytial virus (HRSV) imposes a significant disease burden on infants and the elderly. Intranasal immunization using attenuated live vaccines and certain vector vaccines against HRSV has completed phase II clinical trials with good safety and efficacy.Recombinant protein vaccines for mucosal immunization require potent mucosal adjuvants. Type I interferon (IFN), as a natural mucosal adjuvant, significantly enhances antigen-presenting cell processing and antigen presentation, promoting the production of T and B cells. **Methods:**This study utilized human α2b interferon (IFN-human) and mouse α2 interferon (IFN-mouse) as nasal mucosal adjuvants in combination with fusion protein (F). Intranasal immunization was performed on BALB/c mice to evaluate the immunogenicity of the formulation in vivo. **Results:**Compared to the F protein immunization group, mice in the F + IFN-Human and F + IFN-Mouse experimental groups exhibited significantly increased neutralizing antibody titers and augmented secretion of IFN-γ and IL-4 by lymphocytes,  and both of them could induce the production of high-titer specific IgA antibodies in mice (*p* < 0.001).The F + IFN-Human immunization induced the highest IgG and IgG1 antibody titers in mice; however, the F + IFN-Mouse immunization group elicited the highest neutralizing antibody titers (598), lowest viral loads in the lungs (Ct value of 31), and fastest weight recovery in mice. Moreover, mice in the F + IFN-Mouse immunization group displayed the mildest lung pathological damage (Total score of pathological injury was 2). **Conclusions:**In conclusion, IFN-Mouse, as a mucosal adjuvant for HRSV recombinant protein vaccines, demonstrated superior protective effects in mice compared to IFN-Human adjuvants.

## 1. Introduction

Human respiratory syncytial virus (HRSV) is a single-stranded, negative-sense RNA virus classified into subtypes A and B. It belongs to the family *Pneumoviridae*, genus *Orthopneumovirus* [1]. HRSV is a leading cause of acute lower respiratory tract infections in infants, the elderly, and immunocompromised individuals [2]. Research estimates that in 2019, HRSV caused over 100,000 deaths globally among children under five years old, with more than 45,000 deaths occurring in infants under six months [3]. Severe HRSV bronchiolitis in infancy is a strong risk factor for the development of allergic asthma in early adolescence [4]. As people age, the incidence of underlying diseases increases, and immune function declines, leading to higher morbidity and mortality rates from HRSV in the elderly [5,6]. HRSV infection can also result in diseases such as interstitial myocarditis, epilepsy, or encephalitis [7].

HRSV infection is confined to the respiratory tract and can be transmitted via respiratory droplets or contaminated surfaces. Natural infection does not confer lasting immunity, as nasal mucosal immunoglobulin A (IgA) antibodies rapidly decline, making reinfection common [8]. Therefore, an ideal HRSV vaccine needs to induce the production of mucosal HRSV-specific IgA antibodies. Nasal immunization is less invasive, does not require specialized medical personnel for administration, and is more acceptable to humans, especially children [8,9]. In 2023, two HRSV vaccines and one monoclonal antibody were approved globally, including GSK’s Arexvy vaccine, Pfizer’s Abrysvo vaccine, and Sanofi’s Beyfortus antibody, all targeting the Pre-F antigen. In May 2024, an mRNA vaccine, mRESVIA, was approved by the FDA to protect adults aged 60 and above from RSV infection. Currently, all marketed RSV vaccines are administered via intramuscular injection.

The respiratory mucosa serves as the first barrier against viral invasion. Increasing evidence suggests that immune responses induced by nasal mucosal immunization are superior to those induced by intramuscular injection. Previously, all HRSV live-attenuated vaccines and some viral vector vaccines in clinical trials were administered nasally. However, live-attenuated vaccines carry the risk of virulence reversion, and vector vaccines are affected by pre-existing antibodies in adults, leading to short retention time in the nasal cavity and low levels of induced immune response. Intranasal immunization with recombinant protein vaccines can avoid issues related to virulence reversion and pre-existing antibodies, but due to rapid enzyme degradation and the mucociliary clearance system, recombinant protein vaccines are quickly eliminated and generally have low immunogenicity, failing to produce protective effects [10]. Therefore, effective mucosal adjuvants are essential to enhance the immune response [11]. Mucosal adjuvants not only strengthen the immune response but also modulate its direction and can reduce the required dose.

Mucosal adjuvants can be broadly categorized into toxoid-like adjuvants, polymer adjuvants, pattern recognition receptor agonists, and cytokine adjuvants. Cytokines, known for their role in activating and regulating adaptive immune responses, are widely used as natural endogenous adjuvants. Common cytokines used as mucosal adjuvants include IFN, IL-1, IL-12, IL-15, and TNF [12]. Interferons (IFNs) are divided into type I, type II, and type III based on their source and physicochemical properties. Type I IFNs are primarily involved in antiviral immune responses but can also act as natural adjuvants, significantly enhancing dendritic cell antigen processing and presentation and promoting the generation of follicular helper T cells (Tfh) and memory B cells [13]. Studies have shown that intranasal immunization with murine IFN-α before HRSV infection in neonatal mice can increase nasal mucosal-specific IgA production [14]. IFNs exhibit species specificity; thus, this study uses human α2b (IFN-Human) and murine α2 (IFN-Mouse) IFNs as mucosal adjuvants to investigate the protective effect of intranasal immunization of BALB/c mice with these adjuvants combined with fusion protein (F).

## 2. Materials and Methods

### 2.1. Cells and Animals

A Hep-2 cell line was purchased from ATCC. The HRSV-Long strain is preserved in our laboratory. Female BALB/c mice, aged 6–8 weeks, were purchased from Beijing Vital River Laboratory Animal Technology Co., Ltd., Beijing, China. The animal experiments involved in this study have been approved by the Animal Experiment Ethics Committee of National Institute for Viral Disease Control and Prevention, Chinese Center for Disease Control and Prevention, China CDC (No. 2022017).

### 2.2. Major Instruments and Reagents

A real-time quantitative fluorescence PCR machine was purchased from Bio-Rad, Hercules, CA, USA; a viral nucleic acid extraction kit from Xi’an Tianlong Technology Co., Ltd., Xi’an, China; a fluorescence quantitative PCR kit PrimeScript™ One Step RT-PCR Kit from Takara, Japan; goat anti-mouse IgG antibody from Dakewe Biotech Co., Ltd., Shenzhen, China; IgG1 and IgG2a antibodies from Abcam, Boston, MA, USA; Mouse IFN-γ precoated ELISPOT kit, Mouse IL-4 precoated ELISPOT kit from MabTech, Stockholm, Sweden; PMA + Ionomycin and mouse lymphocyte separation medium from Dakewe Biotech Co., Ltd.; expression vector gene synthesized and cloned into the pCDNA3.4 plasmid by General Biosystems, Chuzhou, China; Top10 strains from Quanshijin Biotechnology Co., Ltd., Beijing, China.

### 2.3. Construction of F Protein and IFN Adjuvants

The pre-fusion F protein (F) is preserved in our laboratory [15]. The F protein sequence is derived from the HRSV A strain (GenBank: KY296733.1). To prevent structural changes after cleavage by furin protease, amino acids 98~144 of F were replaced with a flexible linker. Additionally, to enhance the trimerization of the monomer, the C-terminal transmembrane and intracellular segments were removed and replaced with aβ-folded conformation Foldon sequence derived from the T4 phage. To enhance the stability of the fusion epitope Ø and the protein, six amino acid mutations were introduced. For purification purposes, 6×His and Strep-tag II tags were added at the protein’s C-terminus. After codon optimization, the nucleotide sequence was cloned into the expression vector pcDNA3.4 [16].

The IFN protein sequences are derived from mouse α2 (IFN-Mouse) (GenBank: P01573.1) and humanα2b (IFN-Human) (GenBank: AAP20099.1). To facilitate purification, a 6× His tag was added at the C-terminus of the protein.

### 2.4. Expression and Purification Verification of IFN Adjuvants

The successfully constructed IFN protein gene was inserted into the pcDNA3.4 plasmid to obtain Top10 strains with the correct amino acid sequence. The strains were inoculated into an LB medium containing 50 μg/mL ampicillin and incubated overnight at 37 °C with shaking. The next day, their growth density was measured, and DNA plasmids containing the IFN gene were extracted. The plasmids were transfected into CHO suspension cells at a certain concentration and cultured for 3 days. The cell-free supernatant was collected using centrifugation.

IFN proteins were purified using Ni Sepharose affinity chromatography to obtain the target protein. Using gradient elution, set up different ratios of liquid A and liquid B. First, set the final concentration of imidazole to 60 mM to wash away impurities from the purification column. Then, set the final concentration of imidazole to 100 mM to continue washing away impurities and some loosely bound target protein. Finally, set the final concentration of imidazole to 300 mM and collect the eluent when the UV absorbance reads increase. Label the collected fractions and take samples for SDS-PAGE electrophoresis verification. Choose an appropriate ultrafiltration tube based on the size of the target protein to concentrate it. When concentrated to 1.5 mL, add 10 times the volume of sterile PBS for exchange to wash away residual imidazole, then concentrate to the desired volume. Take samples for SDS-PAGE electrophoresis to verify purity, perform Western blotting to confirm the target band, and finally use the Pierce^TM^ BCA Protein Assay Kit for protein quantification (Pierce, Appleton, WI, USA).

### 2.5. Mouse Immunization Strategy

Select 30 female BALB/c mice aged 6 to 8 weeks and randomly divide them into 5 groups, with 6 mice per group. Blood collection and immunization (50 μL per mouse under isoflurane anesthesia) were performed on day 0 and day 21. Human α2b (IFN-Human) and mouse α2 (IFN-Mouse) interferons were used as mucosal adjuvants. The immunization protocols for each group are detailed in Table 1. The challenge procedure was as follows: on day 35, under isoflurane anesthesia, mice were intranasally infected with 3 × 10^5^ PFU of the HRSV A-long live virus (50 μL per mouse). Mice weight changes were recorded daily, and on day 4 post-challenge, mice were euthanized, and lung tissues were collected.

### 2.6. Sodium Dodecyl Sulfate Polyacrylamide Gel Electrophoresis (SDS-PAGE)

In a 1.5 mL EP tube, add 20 μL of the target protein and 5 μL of 5× Sample Buffer, and boil for 10 min. Insert a 5–20% pre-made gel into the electrophoresis tank, prepare the electrophoresis buffer, and pour it into the tank until the liquid covers the sample wells. Load 5 μL of Protein Marker and 20 μL of the sample into the sample wells. Close the electrophoresis tank lid, connect the power leads, and electrophorese at 200 V for 30–40 min until the bromophenol blue band reaches the bottom of the gel. After electrophoresis, remove the gel and place it in a Coomassie Brilliant Blue staining solution on a shaker for 15–20 min until the background is clear.

### 2.7. Enzyme-Linked Immunosorbent Assay (ELISA)

Each well of a 96-well plate was coated with 50 μg/well of F protein and incubated overnight at 4 °C. The next day, the plate was washed four times with PBST buffer and dried. Each well was blocked with PBS containing 10% FBS and incubated at 37 °C for 2 h. After washing and drying, sera diluted in a 1:100 ratio with five-fold gradient dilutions were added to the wells and incubated at 37 °C for 1 h. After washing and drying, HRP-labeled goat anti-mouse IgG (1:5000 dilution), IgG1 (1:20,000 dilution), and IgG2a (1:2000 dilution) antibodies were added to the wells and incubated at 37 °C for 30 min. After washing and drying, TMB substrate solution was added and incubated at 37 °C for 15 min. Finally, a stop solution was added, and the absorbance (OD value) of each well was measured at 450 nm using a microplate reader. Serum antibody titer (EC50) was obtained by four-parameter curve fitting in GraphPad Prism9 software. The background reading is subtracted from each hole before the curve is fitted.

### 2.8. HRSV Neutralization Assay

Mouse serum was treated at 56 °C for 30 min to inactivate complement. Serum samples were serially diluted two-fold, starting from 1:8 in DMEM containing 2% FBS, with each sample prepared in duplicate to ensure experimental reliability and accuracy. The diluted serum was mixed 1:1 with the HRSV/Long strain (1500 PFU/mL) for a total volume of 100 µL. The virus–serum mixture was incubated at 37 °C and 5% CO₂ for 2 h. After incubation, the mixture was added to 3 × 10^5^ Hep-2 cells per well for 1 h of adsorption. The viral fluid was discarded, and the cells were covered with a 1.2% carboxymethylcellulose medium and placed in a 37 °C incubator. After 4 days, the overlay was removed, and the cells were fixed with 0.04% crystal violet stain for 10 min, followed by rinsing and air-drying. The number of plaques in each well was recorded, and serum-neutralizing antibody titers (IC50) were determined by four-parameter curve fitting in GraphPad Prism 9 software. The HRSV/Long strain used for the neutralization assay and mouse challenge were derived from the same stock.

### 2.9. Enzyme-Linked Immunospot Assay (ELISPOT)

Lymphocytes from the spleen tissues of BALB/c mice were isolated and counted according to the instructions of the mouse lymphocyte separation medium from Dakewe Biotech Co., Ltd. According to the instructions of the ELISPOT kit (MabTech Biotech Co., Ltd., Stockholm, Sweden), cytokines (IFN-γ, IL-4) were detected. The ELISPOT plate was washed four times with sterile PBS. The plate was incubated at room temperature with 1640 medium containing 10% FBS for at least 30 min. The medium was removed, and 100 μL of cells were added to each well, followed by 10 μL of stimulants, including F protein and the positive control phytohemagglutinin (PHA). The plate was incubated in a 37 °C incubator with 5% CO_2_ for 48 h. The liquid was discarded, and the plate was washed five times. Detection antibodies diluted in PBS containing 0.5% FBS were added at 100 μL/well and incubated at room temperature for 2 h. After washing five times, streptavidin-HRP diluted in PBS containing 0.5% FBS was added at 100 μL/well and incubated at room temperature for 1 h. The plate was washed five times, and 100 μL of TMB substrate solution was added to each well. After visible spots appeared, the plate was washed with pure water to stop the reaction. The plate was air-dried in the dark, and the spots were counted using an ELISPOT reader.

### 2.10. Measurement of Viral Load in Mouse Lung Tissues

The right lung tissues of mice were aseptically collected and ground with 2% DMEM medium. A 200 μL sample was taken for nucleic acid extraction, and the viral load of HRSV in the lung tissues was detected using real-time quantitative fluorescence PCR (RT-PCR). The left lung tissues were fixed with 4% paraformaldehyde for 24 h, stained with hematoxylin and eosin (HE), and examined for tissue characteristics, including alveolitis, bronchiolitis, and the extent of inflammatory cell infiltration around blood vessels and in the interstitium [17]. The lung tissues of individual mice were blindly evaluated, and the assessment indicators included alveolar wall thickening, interstitial alveolitis, alveolitis, and bronchiolitis; alveolar septal widening, and inflammatory cell (mononuclear cells, neutrophils, etc.) infiltration around vessels or bronchi were analyzed. Lung pathology was comprehensively scored based on the following criteria: 0 (normal, no lesions), 1 (mild lesions, lesion area less than 1/4 of the lung section), 2 (moderate lesions, lesion area about 1/4 to 2/4 of the lung section), 3 (severe lesions, lesion area about 2/4 to 3/4 of the lung section), 4 (extremely severe lesions, lesion area greater than 3/4 of the lung section).

### 2.11. Statistical Analysis

The data were organized using Excel, and statistical analysis and plotting were performed with GraphPad Prism 9.0. One-way ANOVA with Tukey’s test was applied to assess the significance of differences between groups, and a non-parametric test (Kruskal–Wallis test) followed by Dunn’s multiple comparison test was used for further analysis. Statistical significance was determined by the following *p*-values: * *p* < 0.05, ** *p* < 0.01, *** *p* < 0.001, and **** *p* < 0.0001. Non-significant differences were indicated as “ns.” In this study, the Kruskal–Wallis test was used to analyze the pathological results, and the one-way ANOVA analysis was used to compare the other groups.

## 3. Results

### 3.1. Expression Identification of Two Types of IFN Mucosal Adjuvants

As shown in Figure 1, the sizes of the two purified target proteins (IFN-Human and IFN-Mouse) are approximately 30 kDa.

### 3.2. IFN as a Mucosal Adjuvant Induces Humoral and Cellular Immune Responses in Mice

Using enzyme-linked immunosorbent assay (ELISA), the humoral immune response induced by two mucosal adjuvants, IFN-Human and IFN-Mouse, combined with F protein in BALB/c mice was evaluated. Figure 2A shows the titers of specific IgG antibodies produced in mice after immunization. As shown, the experimental group F + IFN-Human induced higher levels of specific IgG antibodies (EC50,8897) compared to the F + IFN-Mouse group (EC50,2703) and the F protein control group (EC50,1043), with significant statistical differences (*p* < 0.01, *p* < 0.001).

Using a plaque reduction neutralization test, the neutralizing-antibody titers produced in BALB/c mice after secondary immunization and challenge were compared. As shown in Figure 2B, both the F + IFN-Human and F + IFN-Mouse experimental groups induced higher levels of neutralizing antibodies in mice compared to the control groups (F, IFN-Human, IFN-Mouse). The F + IFN-Mouse experimental group induced higher neutralizing-antibody titers than the F + IFN-Human experimental group, with significant statistical differences (*p* < 0.001).

Using the Enzyme-Linked Immunospot (ELISpot) assay, the number of cytokines (IFN-γ, IL-4) secreted by spleen lymphocytes in mice after immunization with different species of IFN mucosal adjuvants was evaluated. As shown in Figure 2C,D, both the F + IFN-Human and F + IFN-Mouse experimental groups secreted higher numbers of IFN-γ cytokines compared to the control groups (F, IFN-Human, IFN-Mouse), with significant statistical differences, but there was no statistical difference between the experimental groups. Both the F + IFN-Human and F + IFN-Mouse experimental groups secreted higher numbers of IL-4 cytokines (F + IFN-Human = 266, F + IFN-Mouse = 195) compared to the control groups (F = 78, IFN-Human = 56), with significant statistical differences (*p* < 0.05, *p* < 0.001), but there was no statistical difference between the experimental groups.

### 3.3. IFN as a Mucosal Adjuvant Induces Polarization of Immune Response in Mice

Figure 3A,B show the titers of specific IgG1 and IgG2a antibodies produced in mice after immunization. As shown in Figure 3A, the experimental groups F + IFN-Human and F + IFN-Mouse induced higher levels of specific IgG1 antibodies compared to the F protein control group (*p* < 0.01), with no statistical difference between the experimental groups. As shown in Figure 3B, the experimental group, F + IFN-Mouse, induced higher levels of specific IgG2a antibodies compared to the F + IFN-Human group and the F protein control group, with a significant statistical difference (*p* < 0.01). Figure 3C shows the ratio of specific antibodies IgG1 to IgG2a produced in mice after immunization. As shown, the experimental group, F + IFN-Human, and the control group F induced higher levels of specific IgG1 antibodies compared to IgG2a; in contrast, the experimental group, F + IFN-Mouse, induced lower levels of specific IgG1 antibodies compared to IgG2a, with a significant statistical difference (*p* < 0.01). This suggests that compared to the control group, both experimental groups F + IFN-Human and F + IFN-Mouse induced higher titers of binding antibodies in mice, with the F + IFN-Mouse group inducing a more Th1-biased humoral response in BALB/c mice. Figure 3D shows the ratio of cytokines IFN-γ to IL-4 secreted by spleen lymphocytes in mice after immunization. There was no statistical difference between the experimental groups F + IFN-Human and F + IFN-Mouse, suggesting that both types of mucosal adjuvants (IFN-Human, IFN-Mouse) induced a balanced cellular immune response.

### 3.4. IgA Antibody Titers, Lung Viral Load, and Body Weight Changes in Mice After Nasal Mucosal Immunization and Challenge

Using an enzyme-linked immunosorbent assay (ELISA), the specific IgA antibody titers in the nasal mucosa of BALB/c mice induced by two mucosal adjuvants, IFN-Human and IFN-Mouse, combined with F protein were evaluated. As shown in Figure 4A, both experimental groups F + IFN-Human and F + IFN-Mouse induced high titers of specific IgA antibodies in mice, with significant statistical differences compared to the control group (F) (*p* < 0.001).

Fourteen days after the second immunization, BALB/c mice were challenged (3 × 10^5^ PFU per mouse), and their body weight was recorded daily. The mice were euthanized aseptically after 4 days, and lung tissues were collected to measure viral lung Ct values. As shown in Figure 4C, the changes in body weight of mice post-infection indicate that on the second day after infection, the body weights of all control groups (F, IFN-Human, and IFN-Mouse) continued to decrease. In contrast, compared to the control group (IFN-Mouse), the experimental groups (F + IFN-Human and F + IFN-Mouse) showed a gradual recovery in body weight, which was statistically significant (*p* < 0.05, *p* < 0.01). Starting from the third day, the body weights of mice in all groups tended to recover, with the recovery rate of the experimental groups (F + IFN-Human and F + IFN-Mouse) being better than that of the control group (IFN-Mouse) (*p* < 0.05).

RT-PCR was used to detect the viral lung load in mice after the challenge. As shown in Figure 4B, the reduction in viral lung load in the experimental groups F + IFN-Human and F + IFN-Mouse was greater than that in the control group (F), with significant statistical differences (*p* < 0.05, *p* < 0.001). Additionally, the reduction in viral lung load in the F + IFN-Mouse experimental group was greater than in the F + IFN-Human experimental group, with a significant statistical difference (*p* < 0.001).

### 3.5. Pathological Damage in Lung Tissue of Mice After Nasal Mucosal Immunization and Challenge

On the fourth day after BALB/c mice were infected with HRSV, their left lungs were collected, stained with hematoxylin and eosin, and analyzed for the degree of lung pathological damage. Pathological scoring was performed as described in the methods. As shown in Figure 5B, the degree of lung pathological damage in the experimental group F + IFN-Mouse (total pathological injury score of 2) was lower than that in the control groups (IFN-Human and IFN-Mouse, with a total pathological injury score of 3.16), with statistical significance (*p* < 0.05). There was no statistical difference between the experimental groups (F + IFN-Human and F + IFN-Mouse).

From the lung pathology sections of mice after the challenge shown in Figure 5A, it is evident that the three control groups showed significant lung pathological damage, with severe alveolar wall thickening and interstitial pneumonia. Although the degree of lung pathological damage was lower in the experimental groups (F + IFN-Human, F + IFN-Mouse), with no statistical difference between the groups, it was visibly apparent that the F + IFN-Mouse group had milder lung pathological damage compared to the F + IFN-Human group, showing only slight inflammatory infiltration and no significant alveolar wall thickening.

## 4. Discussion

The respiratory tract is a complex system that facilitates gas exchange while forming physical and immune barriers between the external environment, blood, and tissue sites. The lungs are frequently exposed to inhaled debris, allergens, pollutants, commensal or pathogenic microorganisms, and respiratory viruses. The SARS-CoV-2 pandemic has brought lung research to the forefront of global attention [18]. Studies have shown that the high rate of breakthrough infections with SARS-CoV-2 vaccines may be due to weak vaccine-mediated upper-respiratory-tract immune responses. This could be improved by intranasal administration of Type I IFN as an adjuvant to existing SARS-CoV-2 vaccines, thereby enhancing its efficacy and durability and providing a first line of defense against respiratory virus infections [19].

Currently, clinical-stage HRSV vaccines include live-attenuated vaccines [20], vector vaccines [21], recombinant protein vaccines [22], and mRNA vaccines [23], most of which are designed based on the F protein. RSV vaccines developed by GSK, Pfizer, and Moderna have completed Phase III clinical trials in elderly populations over 60 years old and have been approved by the U.S. Food and Drug Administration (FDA) for vaccination in adults aged 60 and above. Whether these vaccines can prevent recurrent HRSV infections remains unknown. Additionally, two monoclonal antibodies for the prevention of HRSV infection have been approved for market release, including Palivizumab (Synagis^®^) and Nirsevimab (Beyfortus) [24]. Studies indicate that, compared to Palivizumab, a single dose of Nirsevimab can maintain approximately 10 times higher and longer-lasting neutralizing-antibody levels within one year of administration, with no significant allergic reactions [25], demonstrating good safety [24]. This further suggests that Nirsevimab holds greater potential for preventing RSV-related lower-respiratory-tract diseases in infants and young children. As a respiratory-transmitted virus, an ideal HRSV vaccine should induce not only humoral and cellular immune responses but also mucosal immune responses, particularly specific IgA antibodies.

Type I IFN can enhance the activity and function of immune cells and, as a mucosal adjuvant, can improve the recognition and processing of antigens by mucosal immune cells, thereby enhancing immune responses. Currently, a SARS-CoV-2 recombinant protein vaccine developed by Livzon Biopharmaceuticals based on human Type I IFN-α4a is undergoing Phase III clinical trials [26]. This study used F protein combined with IFN-Mouse/Human adjuvants for intranasal immunization, resulting in high titers of binding and neutralizing antibodies, and induced high titers of specific IgA antibodies in mice, with F + IFN-Mouse inducing a Th1-biased humoral immune response. Therefore, Type I IFN as a mucosal adjuvant can stimulate the body to produce triple immune responses: humoral, cellular, and mucosal immune responses.

Additionally, type I interferon (IFN) can act as an immunomodulator, playing an important role in enhancing both innate and adaptive immune responses [27]. IFN also plays a key role in regulating B cell activation, antibody production, differentiation, and class switching [28]. This study shows that compared to F + IFN-Mouse, F + IFN-Human induces a higher binding-antibody titer (2703 vs. 8897) and a lower neutralizing-antibody titer (598 vs. 87.5). This phenomenon may be related to differences in the way human-derived IFN activates signaling pathways in mice, as compared to mouse-derived IFN [29]. Both human and mouse type I interferon receptors (IFNARs) consist of two subunits, IFNAR1 and IFNAR2, with amino acid homology between human and mouse IFNAR1 at 50.7% and IFNAR2 at 50.4% [30,31]. This difference could affect the binding efficiency of human IFN to mouse receptors, leading to changes in signal strength and pathways. In mice, human-derived IFNs may not fully activate all downstream pathways of mouse IFNAR, or the signaling pathway in B cells might be affected. This receptor difference partly explains the observed differences between binding-antibody responses and neutralizing activity induced by human IFN in mice. Further research is needed to clarify these effects, such as using SARS-CoV-2 spike protein as an immunogen to observe how human and mouse IFNs influence binding- and neutralizing-antibody production.

Despite the promising application prospects of Type I IFN as a mucosal adjuvant, there are still various challenges in practical application. Currently, Type I IFN has been approved for use in several viral infections, but long-term high-dose use can cause toxicity in patients. Additionally, interferon has a relatively small molecular weight, is unstable in vivo, easily degraded by serum proteases, has a short half-life, and requires frequent administration, leading to low patient compliance. Studies have shown that human IFN-α2b has some preventive and therapeutic effects against HRSV infection in both humans and cotton rats, but the impact of dosage on efficacy is significant [17]. After HRSV infection, non-structural proteins NS1, NS2, and attachment protein G can inhibit the production of Type I IFN. Neonatal mice induce significantly lower levels of Type I IFN compared to adult mice, consistent with studies on HRSV infection in humans. Additionally, the presence of Type I IFN is associated with mild symptoms of HRSV infection in infants, and administering Type I IFN before HRSV infection can reduce lung pathological damage in neonatal mice [32].

Combining influenza virus HA antigen with mouse Type I IFN, a single intranasal immunization in mice can induce good protective effects, preventing influenza virus infection and weight loss [33]. Although Type I IFN shows significant adjuvant effects in mice, human trials have not enhanced neutralizing antibody titers, possibly related to the dosage of Type I IFN [34]. This study expressed humanized and mouse-derived IFN adjuvants, with an immunization dose of 5 µg, without using international units. After nasal mucosal immunization of BALB/c mice with F protein combined with IFN adjuvants and subsequent challenge, the experimental group mice showed faster weight recovery, lower viral lung load, and less lung pathological damage compared to the control group. The protective effect of the mouse-derived IFN adjuvant was superior to that of the human-derived IFN adjuvant. This demonstrates that the Type I IFN expressed in this study, as a mucosal adjuvant, has good safety and protective effects in mice. Further studies will continue to evaluate its dosage and safety.

In summary, as mucosal adjuvants, both human-derived and mouse-derived IFNs provide some protection in mice. Compared to the IFN-Human adjuvant, the IFN-Mouse adjuvant better reduces viral lung load and lung pathological damage in mice, with the fastest weight recovery and a Th1-biased immune response. This indicates that the IFN-Mouse adjuvant is more suitable for mice, and it is inferred that the IFN-Human adjuvant may have better effects in humans. This provides a reference value for the application of IFN-Human mucosal adjuvant. Overall, interferon as a mucosal adjuvant still has potential in immunization research and drug development, but more research support and clinical practice are needed to establish its effectiveness and safety.

## Figures and Tables

**Figure 1 vaccines-12-01297-f001:**
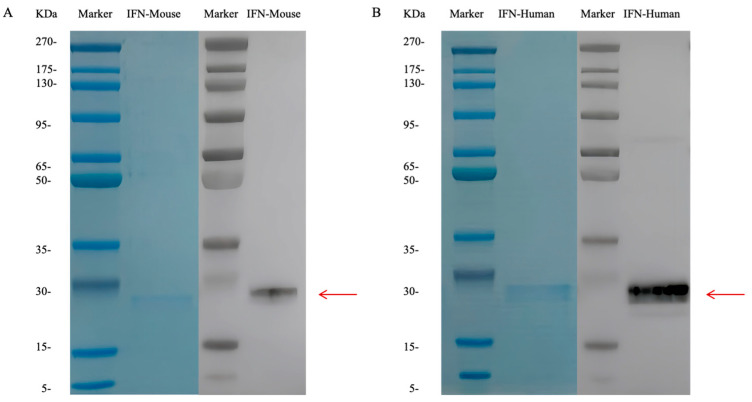
Schematic diagram of HEK293 cells expressing two IFN proteins. SDS-PAGE and Western blot were used to confirm that the protein sizes of (**A**) IFN-Mouse and (**B**) IFN-Human were accurate.

**Figure 2 vaccines-12-01297-f002:**
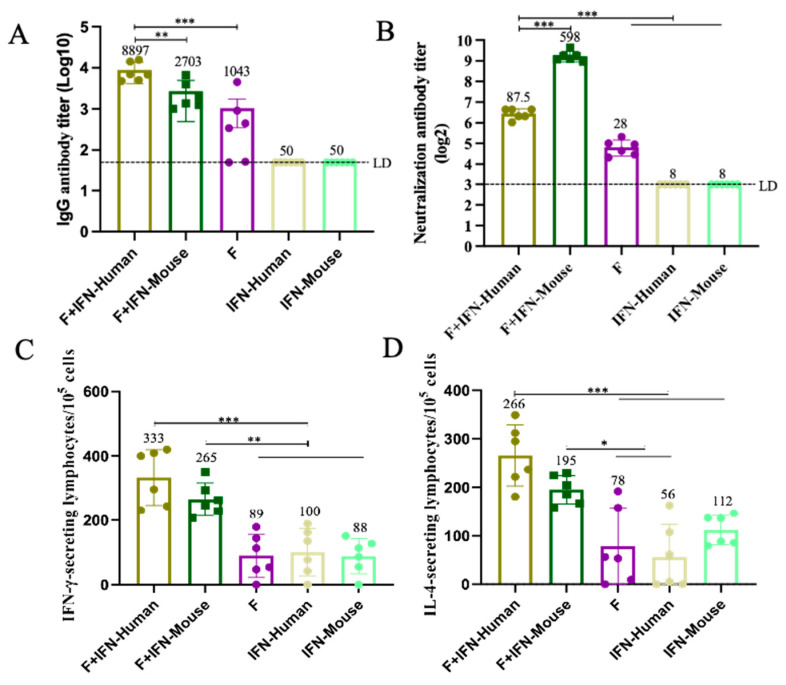
Humoral and cellular immune responses were induced in mice after immunization (**A**): Serum-specific IgG antibody level after immunization. The data represent the geometric means ± SD. LD indicates the limit of detection, which is half of the lowest dilution of serum, and for this experiment, LD = 50. (**B**) The level of serum-neutralizing antibody after immunization: The data represent the geometric means ± SD. LD indicates the limit of detection, which is half of the lowest dilution of serum, and for this experiment, LD = 8. (**C**) The number of cytokines that induce mice spleen lymphocytes to secrete IFN-γ after immunization. (**D**) Cytokine number of IL-4 secreted by spleen lymphocytes of mice after immunization: The data of cytokine secreting cells represent the difference between the number of spots per 3 × 10^5^ cells in the Pre F stimulation hole and the number of spots per 3 × 10^5^ cells in the medium treatment hole. Data represent average ± SD. Statistically significant differences were measured by appropriate one-way ANOVA (* *p* < 0.05; ** *p* < 0.01; *** *p* < 0.001). For the experimental data in this study, each group was set up with replicates to ensure the reproducibility of the results, and the average values were used for subsequent statistical analysis. Data represent two independent experiments.

**Figure 3 vaccines-12-01297-f003:**
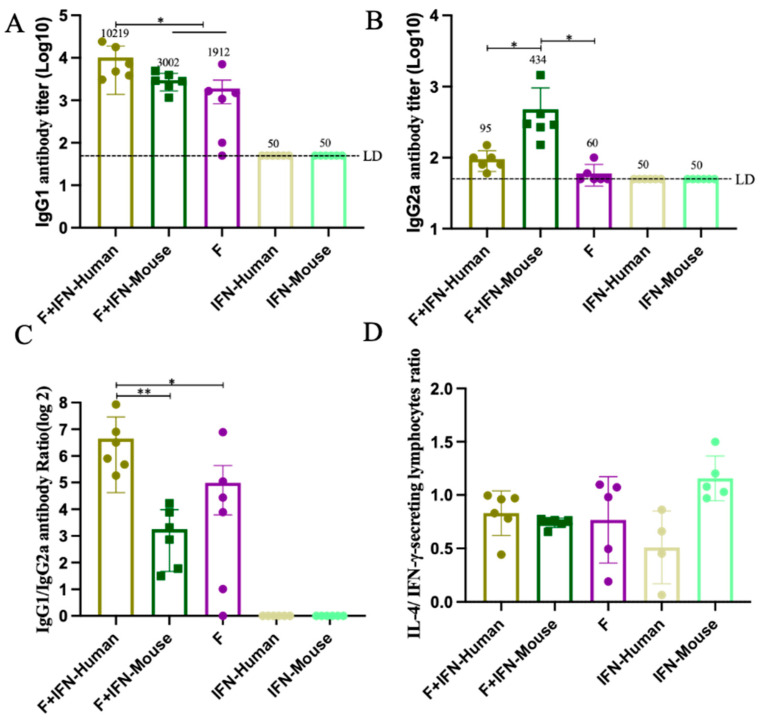
Bias of induced immune response in mice after immunization: (**A**) The antibody level of serum specific IgG 1 after immunization; (**B**) the serum specific IgG 2a antibody level of mice after immunization. The data represent the geometric means ± SD, LD indicates the limit of detection, which is half of the lowest dilution of serum, and for this experiment, LD = 50; (**C**) the ratio of serum-specific antibody IgG 1 to IgG 2a after immunization; (**D**) the ratio of cytokine IL-4/IFN-γ secreted by spleen lymphocytes of mice after immunization; statistically significant differences were measured by appropriate one-way ANOVA (* *p* < 0.05; ** *p* < 0.01).

**Figure 4 vaccines-12-01297-f004:**
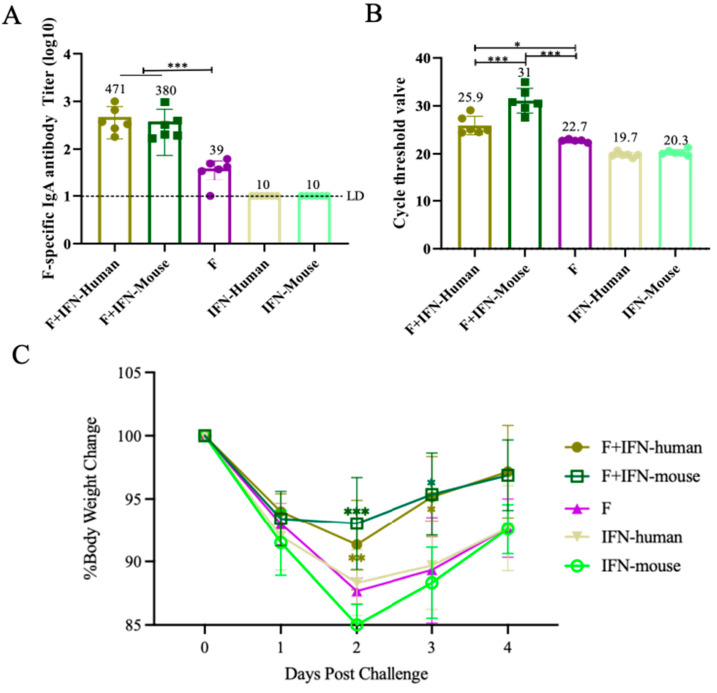
Changes in IgA antibody titers, viral lung load, and body weight in mice after challenge: (**A**) Serum-specific IgA antibody level of mice after 4d challenge. The data represent the geometric means ± SD, LD indicates the limit of detection, which is half of the lowest dilution of serum, and for this experiment, LD = 10; (**B**) CT values of viral lung load of mice after 4d challenge. The data represent the geometric means ± SD; (**C**) changes in body weight of mice after 4d challenge. The average relative body weight ± SEM of all mice in each group (compared with control group); one-way ANOVA analysis was performed using the one-way ANOVA function in GraphPad Prism, followed by Tukey’s multiple comparison test to further analyze the differences between different groups on the same day post-infection (* *p* < 0.05; ** *p* < 0.01; *** *p* < 0.001), and the color indicates the group.

**Figure 5 vaccines-12-01297-f005:**
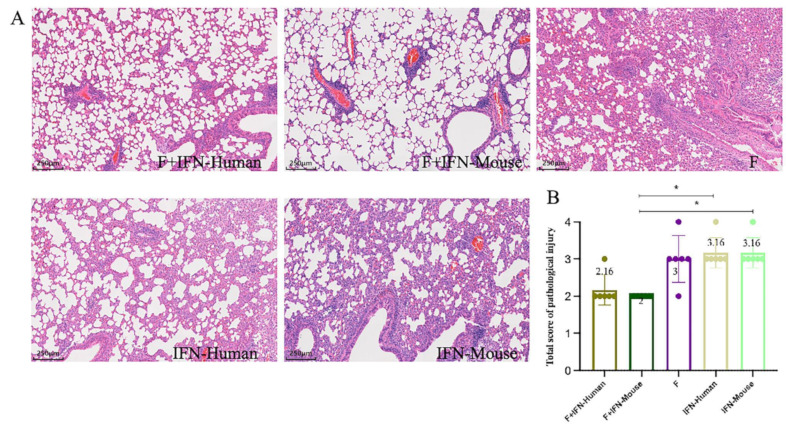
Degree of pathological injury to the lung in mice after challenge: (**A**) pathological sections of mouse lungs after challenge; (**B**) total score of pathological injury to the lungs of mice after challenge, * *p* < 0.05.

**Table 1 vaccines-12-01297-t001:** Immunization protocols for different groups of mice.

Group (*n* = 6)	Antigen	Adjuvant	Route	Day
F + IFN-Human	F	IFN-Human	i.n	0, 21
F + IFN-Mouse	F	IFN-Mouse	i.n	0, 21
F	F	/	i.n	0, 21
IFN-Human	/	IFN-Human	i.n	0, 21
IFN-Mouse	/	IFN-Mouse	i.n	0, 21

(1) The antigen dose was 10 μg of Pre-F protein. (2) The adjuvant used in the intranasal-immunization group was 10 μg IFN-α adjuvant. (3) *n*: Number; i.n.: intranasal; Day: immunization time.

## Data Availability

All the data from the study are available from the corresponding author upon reasonable request.

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
