# Peer review of "Preliminary Study on Type I Interferon as a Mucosal Adjuvant for Human Respiratory Syncytial Virus F Protein"

_vaccines, 2024, doi:10.3390/vaccines12111297_

Round 1
Reviewer 1 Report
Comments and Suggestions for Authors
The study is well-structured and clearly presented. Although it does not use an innovative approach, it is relevant to the field. The methodology is well described and in line with vaccine studies, and the authors explored a large amount of immunological and pathological information in immunized and challenged animals. In this context, I would consider the manuscript for publication.
Reviewer 2 Report
Comments and Suggestions for Authors
The authors investigated the effect of human and mouse IFN-α on the immune response to RSV F-protein. They report that both IFNs enhance the immune response following nasal administration, however, mouse IFN induced a stronger neutralizing antibody response and a greater IgG2a titer.
Methods, line 113-120. The mice were inoculated intranasally with 50 μl of vaccine formulation. This is a very large volume for a mouse. Were the mice anesthetized for the inoculation? What was the volume for the viral challenge?
Methods, line 135. How were antibody titers calculated?
Methods. The authors should describe the method for measuring neutralizing antibodies and the statistical analysis that was performed. The analysis should be performed on log-transformed titers for the comparison of antibody titers and should include a multiple comparison correction.
Figure 2. Please indicate in the legend what the bars and error bars indicate, e.g. mean or geometric mean, SE or SD? Also indicate in the legend what is represented by ** and ***. Same comment for the legends of Figures 3 and 4.
Results, line 218. This should be Figures 3A and B.
Results, line 245. This should be Figure 4A.
Results, line 253-258. Please use statistical analysis to support the stated claims about the bodyweights.
Discussion. Line 314 – 316. “Additionally, Type I IFN has antiviral activity, inhibiting viral replication and proliferation in the respiratory tract, reducing the occurrence and spread of viral infections[17,24], and has direct preventive and therapeutic effects in some infectious and autoimmune diseases16.” This is irrelevant to the discussion of the adjuvant activity of type I IFN as the vaccine is typically administered well before a challenge/natural exposure to the virus.
Discussion. The authors should discuss why human IFN induced a higher systemic antibody titer, but lower neutralizing antibody titers compared with mouse IFN.
Reviewer 3 Report
Comments and Suggestions for Authors
REVIEW
for the manuscript “Preliminary Study on Type I Interferon as a Mucosal Adjuvant
for Human Respiratory Syncytial Virus F Protein
Authors: Hongqiao Hu, Li Zhang, Lei Cao, Jie Jiang, Yuqing Shi, Hong Guo, Yang Wang, Hai Li and Yan Zhang
The title of the article accurately reflects its content.
A brief summary
Aim of the paper was to assess whether interferon Type I (IFN) enhanced as mucosal adjuvants immune response to co-administered intranasally the fusion protein (F) of RSV.
In this study, human α2b (IFN-Human) and mouse α2 (IFN-Mouse) interferons were used as mucosal adjuvants for intranasal immunization of co-administered with the fusion protein (F). Compared to the F protein immunization group, mice in the F+IFN-Human and F+IFN-Mouse experimental groups exhibited significantly increased neutralizing antibody titers and augmented secretion of IFN-γ and IL-4 by lymphocytes. The F+IFN-Human immunization induced the highest IgG and IgG1 antibody titers in mice; however, the F+IFN-Mouse immunization group elicited the highest neutralizing antibody titers, lowest viral loads in the lungs and fastest weight recovery in mice. Moreover, mice in the F+IFN-Mouse immunization group displayed the mildest lung pathological damage. IFN-Mouse as a mucosal adjuvant for HRSV recombinant protein vaccines demonstrated superior protective effects in mice compared to IFN-Human adjuvants.
Contribution: The authors showed that IFN-Mouse as a mucosal adjuvant for RSV recombinant protein vaccines demonstrated superior protective effects in mice compared to IFN-Human adjuvants.
Strengths: The authors experimentally proved that IFNs as adjuvants exhibited species specificity and this will probably need to be taken into account when choosing an adjuvant for RSV vaccines administered intranasally.
General concept
Highlighting areas of weakness:
Introduction.
· Lines 41-42. “In children and adults, HRSV typically leads to upper respiratory tract infections with milder symptoms [5]”. This contradicts the information presented in lines 36-37: “HRSV is a leading cause of acute lower respiratory tract infections in infants, the elderly, and immunocompromised individuals [2]”. To eliminate it, the sentence on lines 41-42 needs to specify the age groups in more detail.
Materials and methods.
• Lines 101-102. Construction of F Protein and IFN Adjuvants is not disclosed. “The pre-fusion F protein (F) is preserved in our laboratory [15]” is indicated only. In my opinion, only the reference is not enough.
• The methods of statistical processing of the research results in the presented article are not specified.
Methodological inaccuracies not identified although some detail in the description of the methods would be useful for reproducing the results in other laboratories;
Missing controls:
• Lines 110-111. “IFN proteins were purified using Ni Sepharose affinity chromatography to obtain the target protein”. In my opinion, it would be useful to mention the methods of controlling the purity of the obtained products.;
Specific comments
Inaccuracies within the text or sentences that are unclear:
· Table 1: It is not clear why the designation F was introduced in the Adjuvant column. The F protein is already indicated in the Antigen column.
· Lines 190, 208, 25 and 274. It is necessary to indicate the numbers of the figures in the text.
Discussion
Line 309. It is advisable to clarify the correctness of the link 23. Authors of this article “sought to engineer a viral antigen that provides greater protection than currently available vaccines and focused on antigenic site Ø, a metastable site specific to the prefusion state of the RSV fusion (F) glycoprotein, as this site is targeted by extremely potent RSV-neutralizing antibodies”. There are many other articles on the effectiveness of using monoclonal antibodies Palivizumab (Synagis®) and Nirsevimab (Beyfortus) in clinical practice.
Line 332. What does the number 16 mean?
The scientific content:
• The manuscript is clear, relevant for the field and presented in a well-structured manner; The authors used F protein combined with IFN- Mouse/Human adjuvants for intranasal immunization, resulting in high titers of binding and neutralizing antibodies, and induced high titers of specific IgA antibodies in mice, with F+IFN-Mouse inducing a Th1-biased humoral immune response. According to the data received, type I IFN as a mucosal adjuvant can stimulate humoral, cellular, and mucosal immune responses. At that, IFNs exhibit species specificity in stimulation of immune response.
• The cited references are mostly relevant but 21 (75%) of the cited 28 publications are beyond the last 5 years; It does not include an excessive number of self-citations;
• The experimental design appropriate to test the hypothesis and, to my opinion, the manuscript can be regarded as scientifically sound. The figures/tables are appropriate and properly show the data.
• Statistical analysis was performed but the methods were not described.
• The conclusion on this work as such and the limitations must be formulated;
Rating the Manuscript
To my opinion, the manuscript is original and well-defined. The results provide an advancement of the current knowledge.
• Scope: The work fits the journal scope.
• Significance: The results mostly interpreted appropriately.
• Quality: The article is written in an appropriate way. The data and analyses presented appropriately.
• Scientific Soundness: The study designed correctly and technically sound. The methods, tools, software, and reagents described without sufficient details to allow another researcher to reproduce the results.
• Interest to the Readers: Article may be of interest for the readership of the journal.
• Overall Merit: The work expands current knowledge on the possibility of increasing the immunogenicity of potential RSV vaccines at intranasal administration of the F protein using host-homologous interferon type 1 as an adjuvant.
• English Level: The English language appropriate and understandable.
• The study reported was carried out in accordance with generally accepted ethical research standards.
Overall Recommendation
The paper can be accepted. Authors are given five days for minor revisions.
Round 2
Reviewer 2 Report
Comments and Suggestions for Authors
The authors’ responses partially addressed my comments. However, several issues remain. In particular, it appears that the authors are confused about the function of type I IFN as adjuvant in their vaccine formulation as they discuss the antiviral effect of type I IFN and antibody responses to IFN (see items 5. and 6. below)
1. The statistical analysis is not correct. The study design has five experimental groups. A one-way ANOVA may reveal that there are statistically significant differences between groups, but it does not indicate which groups are different from each other. For this, a post-hoc test should be performed such as a Bonferroni or Tukey’s with multiple comparison correction. Furthermore, the statistical analysis of the ELISA antibody titers and neutralizing antibody titers should be performed on log-transformed data. The pathology results are based on ordinal data (score from 0 – 4, according to the Methods, lines 216-219) and should be analyzed by a non-parametric test (Kruskal-Wallis followed by a multiple comparison test).
2. The error bars in Figures 2 and 3 should be consistent. Some of the error bars are up/down, others are only up (e.g. group F in Figures 2A, 3A and 3C).
3. Figure 2. Please indicate in the legend whether the data are representative of more than one experiment (in other words, did the authors repeat the mouse experiment with the same outcome?).
4. Figure 4C. It is not clear what the ** indicate. Which groups are statistically significant different from each other? This should be done after a multiple comparison correction as discussed under item 1.
5. Figure 5B. According to the Methods, the scoring scale is 0 – 4. The authors should adjust the Y-axis accordingly.
6. Line 386-391. The antiviral activity of type I IFN is irrelevant when the IFN is used as adjuvant in a preventive vaccine. The protection is based on the induced immune response, not the antiviral activity of the adjuvant. Similarly, lines 421-434 discusses the use of type I IFN as an antiviral therapy and is not relevant to the use of IFN as adjuvant. These sections should be deleted.
7. Lines 397-420. The explanation for the higher antibody titer, but lower neutralizing activity with human IFN as adjuvant does not make sense. The authors discuss human IFN as a foreign protein that is immunogenic. However, they are using human IFN as adjuvant, not as a vaccine antigen. The authors should instead discuss how adjuvants can shape the quality of the immune response in addition to the magnitude of the response. Is there any information on differences in cell signaling between human and mouse IFN with mouse cells?
Round 3
Reviewer 2 Report
Comments and Suggestions for Authors
The authors have adequately addressed my comments.